# Comparison of Fluoroquinolones and Other Antibiotic Prophylaxis Regimens for Preventing Complications in Patients Undergoing Transrectal Prostate Biopsy

**DOI:** 10.3390/antibiotics11030415

**Published:** 2022-03-20

**Authors:** Gabriele Tulone, Sofia Giannone, Piero Mannone, Alessio Tognarelli, Tommaso Di Vico, Rosa Giaimo, Alessandro Zucchi, Marta Rossanese, Alberto Abrate, Nicola Pavan, Francesco Claps, Vincenzo Ficarra, Riccardo Bartoletti, Alchiede Simonato

**Affiliations:** 1Urology Section, Department of Surgical, Oncological and Stomatological Sciences, University of Palermo, 90100 Palermo, Italy; sofia.giannone@gmail.com (S.G.); docpieromannone@gmail.com (P.M.); rosellina.giaimo@libero.it (R.G.); alchiede@gmail.com (A.S.); 2Department of Translational Research and New Technologies in Medicine and Surgery, University of Pisa, 56126 Pisa, Italy; alessio.tognarelli@gmail.com (A.T.); tommaso.divico@yahoo.it (T.D.V.); zucchi.urologia@gmail.com (A.Z.); riccardo.bartoletti@hotmail.com (R.B.); 3Urology Section, Department of Human and Pediatric Pathology “Gaetano Barresi”, University of Messina, 98168 Messina, Italy; marta.rossanese@gmail.com (M.R.); vincenzo.ficarra@unime.it (V.F.); 4Urology Unit, Department of Surgery, ASST Valtellina e Alto Lario, 23100 Sondrio, Italy; alberto.abrate@gmail.com; 5Urology Clinic, Department of Medical, Surgical and Health Sciences, University of Trieste, 34127 Trieste, Italy; nicpavan@gmail.com (N.P.); claps.francesco@gmail.com (F.C.)

**Keywords:** prostate biopsy, antibiotic prophylaxis, fluoroquinolones

## Abstract

Our study aimed to compare the incidence of infective complications after transrectal ultrasound-guided prostate biopsy (TRUSBx) when adopting different antimicrobial prophylaxis regimens. A multi-institutional cohort of 1150 patients who underwent TRUSBx was retrospectively analyzed. Procedures were performed between 2017 and 2019 (before and after the EMA warning about the use of fluoroquinolones for the antibiotic prophylaxis of patient candidates to TRUSBx). The primary endpoint was the occurrence of infective complications, including sepsis and/or fever. The population was stratified according to the antibiotic prophylaxis adopted: fluoroquinolones (levofloxacin, ciprofloxacin, prulifloxacin), cephalosporins (cefixime, ceftriaxone) or trimethoprim/sulfamethoxazole. Univariable and multivariable binomial logistic regression models were used to assess the odds ratio (OR) with 95% confidence interval (CI) testing of the risk of infective complication after adjusting for each prebiopsy covariate. In total, 478 (41.6%) patients received fluoroquinolone-based prophylaxis. Among these, 443 (38.5%), 25 (2.2%) and 10 (0.9%) patients received levofloxacin prophylaxis, ciprofloxacin and prulifloxacin, respectively while 14.6% received cefixime, 20.7% received the comedication of ceftriaxone/fosfomycin and 23.1% received trimethoprim/sulfamethoxazole. The trimethoprim/sulfamethoxazole and fluoroquinolone regimens were significantly associated with a lower risk of infective complications (OR 0.15, 95% CI 0.03–0.48, *p* = 0.003 and OR 0.17, 95% CI 0.06–0.43, *p* < 0.001, respectively). The ceftriaxone/fosfomycin (OR 0.21, 95% CI 0.04–0.92, *p* = 0.04) and fluoroquinolone (OR 0.07, 95% CI 0.00–0.70, *p* = 0.048) prophylaxis were associated with a lower risk of infective sequelae. Fluoroquinolone-based prophylaxis was associated with a lower risk of infective complications after TRUSBx compared to other prophylaxis regimens although its clinical application was recently forbidden by European Medical Agency restrictions.

## 1. Introduction

Antibiotic prophylaxis for prostate biopsy remains a critical issue because of the importance of preventing infectious complications [1]. Infectious complications following prostate biopsy are increasing, and the use of fluoroquinolone prophylaxis has recently been limited by the European Medicines Agency (EMA) [2,3]. The incidence of infectious complications was shown to be significantly higher following transrectal ultrasound-guided prostate biopsy (TRUSBx) compared to transperineal biopsy (relative risk, RR: 1.81 (range 1.09–3.00)) [4,5]. In addition, a meta-analysis including 165 studies with 162,577 patients reported sepsis rates of 0.1% versus 0.9% for transperineal and transrectal biopsies, respectively [6]. Although the European Association of Urology (EAU) guidelines recently recommended transperineal prostatic biopsies, many centers still perform TRUSBx because urologists seem more confident with this practice [7,8]. Complications of TRUSBx, such as hematospermia, hematuria and rectal bleeding, are common but typically mild [9]. Severe complications, such as complicated urinary tract infections (UTIs) and sepsis, can occur, but, fortunately, they are rare. Sepsis is one of the most feared complications of TRUSBx, with an incidence of 1–3% [10]. To minimize severe complications, antibiotic prophylaxis is recommended [11]. Antimicrobial prophylaxis is not standardized, and prolonged prophylaxis is common [12]. The guidelines recommend other hygienic measures, such as rectal cleansing with povidone-iodine before the procedure. However, there is an increased incidence of postbiopsy sepsis and UTIs, probably due to the rise of bacterial antibiotic resistance [1,10,13]. Recently, fluoroquinolones were indicated by the EAU and American Urology guidelines as the first choice for prophylaxis in TRUSBx [14,15,16] due to their tissue tropism. However, in 2018, the EMA limited the use of this class of drugs in patients at risk of aortic aneurysms. In March 2019, the European Commission (EC) implemented stringent conditions regarding the use of fluoroquinolones as a consequence of proven long-lasting, disabling and potentially irreversible adverse drug reactions associated with their use (European Commission final decision, EMEA/H/A-31/1452) [17]. The EC banned fluoroquinolones and recommended the use of fosfomycin trometamol, cephalosporins and aminoglycosides for antimicrobial prophylaxis in patient candidates for TRUSBx [18]. Our study aimed to compare the incidence of complications after TRUSBx with the use of different classes of antibiotics for prophylaxis.

## 2. Results

Data for 1150 patients who underwent TRUSBx between January 2017 and December 2019 were collected and analyzed. The patients had a median age of 70 years (IQR 64–76) and a median prebiopsy PSA of 7.36 ng/mL (IQR 5.4–11.0). In our cohort, 143 (12.4%) patients had well-controlled diabetes. The median number of biopsy cores was 12 (IQR 12–16). Overall, 478 (41.6%) patients received fluoroquinolone-based prophylaxis. Among these, 443 (38.5%), 25 (2.2%) and 10 (0.9%) of patients received levofloxacin, ciprofloxacin or prulifloxacin, respectively. Other patients received cefixime (14.6%), ceftriaxone/fosfomycin (20.7%) or trimethoprim/sulfamethoxazole (23.1%) (Table 1).

The total number of infective complications was 37 (28 fever and 9 sepsis). Among the overall cohort, in the univariable binomial logistic regression, the number of biopsy cores and the presence of diabetes were associated with an increased risk of infective complications (OR 1.42, 95% CI 1.17–1.78, *p* < 0.001 and OR 5.82, 95% CI 2.72–12.2, *p* < 0.001, respectively) (Table 2).

Conversely, trimethoprim/sulfamethoxazole and fluoroquinolones were significantly associated with a lower risk of infective complications (OR 0.15, 95% CI 0.03–0.48, *p* = 0.003 and OR 0.17, 95% CI 0.06–0.43, *p* < 0.001, respectively) (Table 3). 

However, in multivariable analysis, fluoroquinolones reduced the risk of infection (OR 0.22, 95% CI 0.05–0.83, *p* = 0.02) while diabetic patients had a higher risk of contracting infections (OR 6.86, 95% CI 3.11–15.0, *p* < 0.001) (Table 3). We evaluated all the biopsy-related complications of the patients. The most common was hematospermia followed by hematuria. Postoperative fever was reported in 2.4% of patients, and only nine (0.8%) patients had sepsis (Table 1).

## 3. Discussion

The results of this retrospective cohort analysis confirmed that fluoroquinolones, in comparison with other antibiotic compounds, remain attractive for antibiotic prophylaxis in patient candidates for TRUSBx. The study design present a limitation because there is no prospectively randomized component. In our results, insignificant geographical differences in antibiotic resistance patterns to fluoroquinolones were identified between each of the involved study centers, and we recorded efficient but suboptimal efficacy of combined cephalosporin/fosfomicin therapy compared to fluoroquinolone administration [2,6]. In the univariate analysis, there was no correlation between the risk of infection after prostate biopsy and the age of the patient.

In recent years, there has been increased risk of antibiotic resistance, e.g., to fluoroquinolones, and many scientific efforts now aim at researching new antibiotics [19]. This article hosted an important discussion on the possibility of reintroducing drugs, such as quinolones, that have been banned by the EMA in recent years.

Patients who received fluoroquinolone prophylaxis in this study had reduced infectious complications postTRUSBx compared to other antibiotic regimens. The study started before the EMA restrictions for fluoroquinolones were considered as stringent for the prophylaxis of patient candidates for prostate biopsy, except for rare cases of subjects with concomitant increased risk of complications due to aortic aneurism. 

When comparing the common cephalosporin (Cefixime), we can say that trimethoprim and fluoroquinolones are 6.7 and 5.8 times more protective, respectively; this changes the multivariate analysis, where trimethoprim leaves the scene completely and only quinolones remain, which are always five times higher than cephalosporin in terms of protection from infections.

In the univariate analysis, there was a correlation between the number of biopsy samples and the risk of infection (1.42 times).

Our findings present fluoroquinolones as an appropriate and superior alternative to other prophylaxis antibiotics used in previous studies [20,21]. In the literature, there are conflicting opinions on the efficacy of various antibiotic prophylaxes before TRUSBx due to the large number of studies published after the EMA note and the new indications given in the guidelines [18]. 

We did not study UTIs because, in most cases, urinary infections were treated by the GP, meaning we could not trace the data of interest. This must be considered a limitation of our study.

A recent, large, Canadian, nested, case-control study with more than 9000 patients reported that fosfomycin (single dose/two doses) was inferior to ciprofloxacin (three days/single dose) for preventing infective complications. The results led to limited fosfomycin administration for antibiotic prophylaxis in Canadian patient candidates for TRUSBx [18].

In the paper published by Qiao et al., the authors reported good results in terms of infection prevention using levofloxacin 500 mg e.v. for three days as a prophylactic regimen, underlining the low cost and economic advantage of this treatment.

Several meta-analyses and RCTs confirmed the greater efficacy of fosfomycin [22,23,24,25]. A 2017 review and meta-analysis, before the EMA note, described fosfomycin’s greater effectiveness in the prevention of infectious complications compared to fluoroquinolones [26]. Ongün et al. [27] demonstrated a lower incidence of urinary tract infections in patients receiving fosfomycin prophylaxis compared to fluoroquinolones. Seyed [28] and colleagues investigated the combination of aminoglycosides and oral antibiotics for three days along with povidone-iodine rectal cleansing and concluded that this may represent an optimum strategy to minimize the risk after TRUSBx. Derin [29] described important repercussions, such as an increased incidence of infectious complications with limited use of fluoroquinolones as the first-choice antibiotics for prophylaxis in prostate biopsies. It is, therefore, necessary to really evaluate the cost−benefit ratio to avoid a delay in surgical treatment with radical prostatectomy due to the sequelae of prostate biopsy. It could be safer to stratify patients according to the risk of infectious complications. This stratification, however, will have to take into account geographical influences and varying antibiotic resistance levels [29]. 

A Norwegian registry study reported increased resistance to trimethoprim/sulfamethoxazole (from 35 to >60%) and fluoroquinolones (from 15 to 45%) from 2013 to 2016, resulting in increased hospitalization (1.5–10%) and relative mortality rates [30]. *The same results were found in Eastern European countries and in Sweden* [31,32]. Additionally, fluoroquinolones and concomitant extended-spectrum β-lactamase (ESBL) production may be as high as 70% and 60%, respectively, in Asian countries, as well as 57% in African countries [33,34]. 

The guidelines now recommend the transperineal technique for prostate biopsy as it certainly presents lower infectious risks than the transrectal technique, which is a messy procedure by definition [18], although the transperineal approach may induce other relevant adverse events, such as persistent perineal pain and local hematomas [35]. This strong indication also coincides temporally with the note of the EMA, a coincidence that could influence the results of antibacterial prophylaxis strategies with a bias related to the different biopsy techniques used. 

## 4. Materials and Methods

A multicentric, retrospective study was performed, collecting the data of patients who underwent TRUSBx between 2017 and 2019 (before and after the publication of the EMA note) in three different centers: Palermo, Pisa and Cuneo. All patients underwent TRUSBx for a strong suspicion of prostate cancer (PCa) based on a PSA and digital rectal examination. Not all patients followed the guidelines of undergoing multiparametric magnetic resonance imaging (mpMRI) before the biopsy [36]. All immediate and delayed adverse events were registered. The exclusion criteria of the study were as follows: patients with antibiotics allergies, positive urine culture or other dual antiplatelet therapies. We analyzed the rates of complications and unplanned visits or readmission after TRUSBx. The primary endpoint was the occurrence of infective complications, including sepsis and/or fever (body temperature ≥ 37.5 °C). Sepsis was defined as life-threatening organ dysfunction caused by a dysregulated host response to infection. The clinical criteria for sepsis include suspected or documented infection and an acute increase of two or more Sequential Organ Failure Assessment (SOFA) points as a proxy for organ dysfunction [37].

### 4.1. Statistical Analysis

Descriptive analysis included frequencies and proportions for categorical variables. Medians and interquartile ranges (IQR) were reported for continuous coded variables. The population was stratified according to the antibiotic prophylaxis adopted: fluoroquinolones (levofloxacin, ciprofloxacin, prulifloxacin), cephalosporins (cefixime, ceftriaxone) or trimethoprim/sulfamethoxazole. The Mann–Whitney U or Kruskal–Wallis test and the chi-squared or Fisher’s exact test were used for comparison of the continuous and categorical data, respectively. All tests were two-sided, with the level of significance set to *p* < 0.05. To test the risk of infective complications, uni- and multivariable binomial logistic regression models were used to assess the odds ratio (OR), with a 95% confidence interval (CI), after adjusting for each prebiopsy covariate, including the following: the age at biopsy, year of biopsy, number of cores retrieved, antibiotic prophylaxis regimen adopted and presence of diabetes. For comparison purposes, we considered several references among categorical variables evaluated in the logistic regression models. These included the adoption of Cefixime as an antibiotic prophylaxis regimen, the absence of diabetes and the 2018 year of TRUSBx, respectively. After univariable analysis factors with *p* < 0.2 were entered into the multivariable model, this was followed by backward elimination to determine the factors most associated with infective complications’ occurrence. Sensitivity analysis on the subgroup of patients with diabetes was performed. Statistical analyses were performed using RStudio v. 1.2.5001 (RStudio: Integrated Development for R, RStudio, Inc., Boston, MA, USA).

### 4.2. Antibiotic Prophylaxis, Patient Preparation and Technique of TRUSBx

All patients followed a rigid schedule of antibiotic prophylaxis the night before and on the day of the procedure. All antibiotics were administered orally except for ceftriaxone, which was administered intramuscularly (Table 4).

An enema was carried out the night before and the morning of the procedure as well as rectal cleansing with povidone-iodine immediately before the biopsy. Patients were usually placed in the left lateral decubitus position with their knees and hips flexed at 90 °; just a few cases were performed in the lithotomy position. A spring-driven 18 G needle core biopsy device was used. Local analgesia was achieved with a transrectal, ultrasound-guided injection of 1% or 2% lidocaine at the junction between the seminal vesicle and the prostate gland and on the apex of the prostate. All patients underwent a cognitive sextant TRUSBx, with 12 cores, plus other cores based on mpMRI findings or ultrasound suspicion. The number of biopsy cores taken also varied in relation to the age of the subject and bleeding. Antiplatelet therapies for primary prevention were stopped at least five to seven days before the biopsy; target therapies were stopped 48 h before the procedure and were readministered the same day of the biopsy. A prebiopsy urine culture was obtained from all the patients three days before the procedure. 

## 5. Conclusions

In conclusion, fluoroquinolone-based prophylaxis appears to be associated with a lower risk of infective complications after TRUSBx compared to other prophylaxis regimens although the European Medical Agency restrictions forbid the prophylactic use of FQs for prostate biopsy. 

## Figures and Tables

**Table 1 antibiotics-11-00415-t001:** Descriptive baseline characteristics and clinicopathological features for the cohort of 1150 patients who underwent transrectal ultrasound prostate biopsy.

Patients, *n* (%)	1150 (100.0)
Age, median (IQR)	70 (64–76)
Prebiopsy PSA (ng/mL), median (IQR)	7.4 (5.4–11.0)
N. of cores, median (IQR)	12 (12–16)
Year of prostate biopsy, *n* (%)	
2018	227 (19.7)
2019	726 (63.1)
2020	197 (17.1)
Antibiotic prophylaxis regimen, *n* (%)	
Cefixime 400 mg	168 (14.6)
Ceftriaxone 1 g/Fosfomycin 3 g	238 (20.7)
Trimethoprim 160 mg/Sulfamethoxazole 800 mg	266 (23.1)
Ciprofloxacin 500 mg	25 (2.2)
Levofloxacin 500 mg	443 (38.5)
Prulifloxacin 600 mg	10 (0.9)
Diabetes, *n* (%)	143 (12.4)
Complications, *n* (%)	
Hematospermia	126 (11.0)
Hematuria	126 (11.0)
Rectal bleeding	86 (7.5)
Fever (>37.5 °C)	28 (2.4)
Sepsis	9 (0.8)
Urinary retention	17 (1.5)
Rectal pain	73 (6.3)
Unplanned visit, *n* (%)	29 (2.5)
Unplanned readmission, *n* (%)	10 (0.9)
Center, *n* (%)	
(Center A)	130 (11.3)
(Center B)	621 (54.0)
(Center C)	399 (34.7)

Abbreviations: IQR, interquartile range; PSA, prostate-specific antigen.

**Table 2 antibiotics-11-00415-t002:** Descriptive baseline characteristics and clinicopathological features for the cohort of 1150 patients who underwent transrectal ultrasound prostate biopsy, according to diabetes status.

Variable	Diabetes		*p*
	no	yes	
Patients, *n* (%)	1007 (87.6)	143 (12.4)	
Age (years), median (IQR)	70 (64–75)	70 (65–76)	0.25
N. of cores, median (IQR)	12 (12–16)	12 (12–16)	0.86
Year of prostate biopsy, *n* (%)			
2018	203 (20.2)	24 (16.8)	0.25
2019	638 (63.4)	88 (61.5)	
2020	166 (16.5)	31 (21.7)	
Antibiotic prophylaxis regimen, *n* (%)			0.03
Cefixime 400 mg	153 (15.2)	15 (10.5)	
Ceftriaxone 1 g/Fosfomycin 3 g	207 (20.6)	31 (21.7)	
Trimethoprim 160 mg/Sulfamethoxazole 800 mg	239 (23.7)	27 (18.8)	
Ciprofloxacin 500 mg	17 (1.7)	8 (5.6)	
Levofloxacin 500 mg	382 (37.9)	61 (42.7)	
Prulifloxacin 600 mg	9 (0.9)	1 (0.7)	
Complications, *n* (%)			
Hematospermia	107 (10.6)	19 (13.3)	0.42
Hematuria	105 (10.4)	21 (14.7)	0.17
Rectal bleeding	74 (7.4)	12 (8.4)	0.78
Fever (>37.5 °C)	15 (1.5)	13 (9.1)	<0.001
Sepsis	6 (0.6)	3 (2.1)	0.16
Urinary retention	14 (1.4)	3 (2.1)	0.78
Rectal pain	55 (5.5)	18 (12.6)	0.002
Unplanned visit, *n* (%)	25 (2.5)	4 (2.8)	0.9
Urinary retention	14	3	
Hematuria	10	1	
Rectal bleeding	1		
Unplanned readmission, *n* (%)	7 (0.7)	3 (2.1)	0.23
Sepsis	6	3	
Hematuria	1		

Abbreviations: IQR, interquartile range; PSA, prostate-specific antigen.

**Table 3 antibiotics-11-00415-t003:** Univariable and multivariable binomial logistic regression analysis for the prediction of infective complications among 1150 patients who underwent transrectal ultrasound prostate biopsy.

Variable	OR (95% CI)	*P*	OR (95% CI)	*p*
Age (years), as cont.	0.98 (0.93–1.02)	0.3	-	-
N. of cores, as cont.	1.42 (1.17–1.78)	<0.001	1.17 (0.93–1.62)	0.3
Year of prostate biopsy				
2018	1.00 (Ref.)	-		
2019	2.66 (0.92–11.3)	0.11	-	-
2020	0.77 (0.10–4.67)	0.8		
Antibiotic prophylaxis regimen				
Cefixime	1.00 (Ref.)	-	1.00 (Ref.)	-
Ceftriaxone/Fosfomycin	0.51(0.20–1.24)	0.14	0.43 (0.17–1.07)	0.07
Trimethoprim/Sulfamethoxazole	0.15 (0.03–0.48)	0.003	0.23 (0.04–1.04)	0.06
Fluoroquinolones	0.17 (0.06–0.43)	<0.001	0.22 (0.05–0.83)	0.02
Diabetes				
No	1.00 (Ref.)	-	1.00 (Ref.)	-
Yes	5.82 (2.72–12.2)	<0.001	6.86 (3.11–15.0)	<0.001

Abbreviations: OR, odds ratio; CI, confidence interval.

**Table 4 antibiotics-11-00415-t004:** Dose, mode and time of administration of antibiotic prophylaxis.

Type of Antibiotic	Mode of Administration	Time of Administration
Cefixime 400 mg	Oral	The night before and the morning of the procedure.
Ceftriaxone 1 g/Fosfomycin 3 g	Ceftriaxone: Administration within a muscle. Fosfomycin: Oral administration	Ceftriaxone: one shot before the biopsy. Fosfomycin: The night before and the morning of the procedure
Trimethoprim 160 mg/Sulfamethoxazole 800 mg	Oral	The night before and the morning of the procedure.
Ciprofloxacin 500 mg	Oral	The night before and the morning of the procedure.
Levofloxacin 500 mg	Oral	The night before and the morning of the procedure.
Prulifloxacin 600 mg	Oral	The night before and the morning of the procedure.

## Data Availability

Data available upon request

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
