# Peer review of "Comparison of Fluoroquinolones and Other Antibiotic Prophylaxis Regimens for Preventing Complications in Patients Undergoing Transrectal Prostate Biopsy"

_antibiotics, 2022, doi:10.3390/antibiotics11030415_

Round 1
Reviewer 1 Report
The authors in this study evaluated the efficacy of various antibiotics regimens by comparing the incidence of infective complications after transrectal ultrasound-guided prostate biopsy via retrospective cohort study design. The findings are conclusive and good to publish.
Levofloxacin was the most frequent among fluoroquinolones-based regimens. Will there be any difference in the efficacy of other fluoroquinolones? Can authors discuss the efficacy of other fluoroquinolones like ciprofloxacin for the prevention? Will it be the same or comparable or do they recommend only levofloxacin as the best option. I know it is a retrospective study and authors can only present what they found in the data but what is their opinion or what they can discuss from previous literature published in this regard
Author Response
Editor
Lois Liu
Assistant Editor
Dorothy Liu
ANTIBIOTICS
Palermo, Feb 28th 2022
Manuscript #: ANTIBIOTICS-1597891
Title: COMPARISON BETWEEN FLUOROQUINOLONES AND OTHER ANTIBIOTIC PROPHYLAXIS REGIMENS IN PREVENTING INFECTIVE COMPLICATIONS IN PATIENTS UNDERGOING TRANSRECTAL PROSTATE BIOPSY
Dear Editors,
Thanks for considering our manuscript for publication in the ANTIBIOTICS. We appreciated the comments made by the reviewers and we prepared a revised version of our manuscript based on these comments. Please also find a point-by-point “response to reviewers” below. We hope you can find this revised version of our manuscript fully suitable for publication in your esteemed journal.
Kindest regards
Gabriele Tulone, MD
Reviewer: 1
The authors in this study evaluated the efficacy of various antibiotics regimens by comparing the incidence of infective complications after transrectal ultrasound-guided prostate biopsy via retrospective cohort study design. The findings are conclusive and good to publish.
Levofloxacin was the most frequent among fluoroquinolones-based regimens. Will there be any difference in the efficacy of other fluoroquinolones? Can authors discuss the efficacy of other fluoroquinolones like ciprofloxacin for the prevention? Will it be the same or comparable or do they recommend only levofloxacin as the best option. I know it is a retrospective study and authors can only present what they found in the data but what is their opinion or what they can discuss from previous literature published in this regard
Thanks for your review. It could be very useful to study the effectiveness of the different quinolones but unfortunately the sample is very small for prulifloxacin and ciprofloxacin so we preferred to evaluate the entire class of quinolones. A future study can be plan to check for any differences.
Reviewer 2 Report
The manuscript submitted by Tulone et al., entitled "Comparison between Fluoroquinolones and other Antibiotic Prophylaxis Regimens in Preventing Infective Complications in Patients Undergoing Transrectal Prostate Biopsy," is quite interesting. The subject is interesting, and presenting more clinical data will improve the prophylactic strategy for prostate biopsy.
The manuscript is not well written and organized, and the English need significant improvements before it can be processed further.
Here are my comments:
- Please delete from abstract- background, methods, conclusions.
- Introduction:
- l. 53-55 references 2 and then 8-9?? Please check the number of references through the whole manuscript
- l. 60-62 needs a reference
- l. 65 [ref]????
- l.67-69 reference 12 is inadequate; the authors refer to EAU and AUA guidelines
- l. 75-78 hard to understand
- Please include the following articles in the Introduction and Discussion section
- https://doi.org/10.3389/fsurg.2018.00002
- https://doi.org/10.1186/s12894-020-00592-8
- https://doi.org/10.1186/s12301-020-00026-9
- 10.12688/f1000research.19260.1
- Results
- Include the tables in the Results section.
- Table 1 - Complications - no UTIs?
- Please detail the reasons for unplanned visits and unplanned readmission in the result section.
- Discussion should first summarize one or two key findings of the study. Why are those findings important in light of previous observations? How we can use these findings in developing antibiotic prophylaxis for prostate biopsy. Do not present at the beginning of the section some limitations.
- l.132-134 needs some references
- l. 161 - 166 the authors should present recent data, especially from Europe:
https://doi.org/10.31925/FARMACIA.2021.3.16
https://doi.org/10.3390/microorganisms8060848 - l. 167-170, the information presented is not correct. The EAU guidelines on prostate cancer - ``Ultrasound (US)-guided biopsy is now the standard of care. Prostate biopsy is performed by either the transrectal or transperineal approach``.
- Please include the limitations of the study.
- Materials and methods
- l. 180 Please specify the centers
- l. 183 needs a reference
- describe clearly the dose of antibiotics, mode, and time of administration
- l. 213-216 hard to understand, please describe ``a rigid scheme of antibiotic prophylaxis``
- Please specify if the patients had enema the night before or on the morning of the procedure
- Conclusions need to be revised
- ``In diabetic patients the different pharmacological efficacy against the infectious risk is probably caused by the different pathogens pharmacological resistance.`` is based on what results???
- Please arrange the whole manuscript and review the references according to journal requirements.
Author Response
Editor
Lois Liu
Assistant Editor
Dorothy Liu
ANTIBIOTICS
Palermo, Feb 28th 2022
Manuscript #: ANTIBIOTICS-1597891
Title: COMPARISON BETWEEN FLUOROQUINOLONES AND OTHER ANTIBIOTIC PROPHYLAXIS REGIMENS IN PREVENTING INFECTIVE COMPLICATIONS IN PATIENTS UNDERGOING TRANSRECTAL PROSTATE BIOPSY
Dear Editors,
Thanks for considering our manuscript for publication in the ANTIBIOTICS. We appreciated the comments made by the reviewers and we prepared a revised version of our manuscript based on these comments. Please also find a point-by-point “response to reviewers” below. We hope you can find this revised version of our manuscript fully suitable for publication in your esteemed journal.
Kindest regards
Gabriele Tulone, MD
Reviewer: 2
The manuscript submitted by Tulone et al., entitled "Comparison between Fluoroquinolones and other Antibiotic Prophylaxis Regimens in Preventing Infective Complications in Patients Undergoing Transrectal Prostate Biopsy," is quite interesting. The subject is interesting and presenting more clinical data will improve the prophylactic strategy for prostate biopsy.
Here are my comments:
- We delete from the abstract background, methods and conclusion
- Introduction:
- 60-62 needs a reference:
Tahnks for your comment. We added the follow references:
[13] Aron, M., et al. Antibiotic prophylaxis for transrectal needle biopsy of the prostate: a randomized controlled study. BJU Int, 2000. 85: 682.
[9]. Abdelkhalek, M.; Abdelshafy, M.; Elhelaly, H.; et al. Hemosepermia after transrectal ultrasound-guided prostatic biopsy: A prospective study Urol Ann. 2013 Jan-Mar
l.67-69 reference 12 is inadequate; the authors refer to EAU and AUA guidelines
Thanks, we refer to the EAU and AUA guidelines before the EAU restriction
[13] Aron, M., et al. Antibiotic prophylaxis for transrectal needle biopsy of the prostate: a randomized controlled study. BJU Int, 2000. 85: 682.
- 75-78 hard to understand
We reformulated the sentence:
The aim of our study is to compare the incidence of complications after TRUSBx. In particular, we reported the events in relation of the use of different class of antibiotics prophylaxis.
Please include the following articles in the Introduction and Discussion section
Thanks a lot for the advice and for the interesting article that we included in the manuscript (Ref. 8, 12, 31)
Results
Include the tables in the Results section
We included table in the Results
Table 1 - Complications - no UTIs?
Thanks for the observation. We could not include the UTIs in the complications because in most of the cases urinary infections were treated by the GP and we were not able to collect the data of interest. We added this as a limitation of our study
Please detail the reasons for unplanned visits and unplanned readmission in the result section.
Thanks for your suggestion. We detailed the reason for unplanned visits e readmission in the table
- Discussion should first summarize one or two key findings of the study. Why are those findings important in light of previous observations? How we can use these findings in developing antibiotic prophylaxis for prostate biopsy. Do not present at the beginning of the section some limitations.
Thanks for you guidance. We refolmulated the paraghraph as it follows.
In recent years there has been an increase in antibiotic resistance, like fluoroquinolones, and many scientific efforts are aimed at researching new antibiotics. This article opens an important discussion on the possibility of reintroducing drugs such as quinolones that have been banned by the EMA in recent years.
l.132-134 needs some references
Ref 16 Pilatz, A.; Veeratterapillay, R.; Dimitropoulos, K.; et al. European Association of Urology Position Paper on the Prevention of Infectious Complications Following Prostate Biopsy, European Urology, 2019
- 161 - 166 the authors should present recent data, especially from Europe:
https://doi.org/10.31925/FARMACIA.2021.3.16
https://doi.org/10.3390/microorganisms8060848
Thanks, we include in the manuscript Ref 30 – 31
30)Chibelean, C. B.; Petca, R.C.; et al.A Clinical Perspective on the Antimicrobial Resistance Spectrum of Uropathogens in a Romanian Male Population Mares Microorganisms, June 2020
31)Styrke, J.;, Resare, S.; Lundström, K.J.; Current routines for antibiotic prophylaxis prior to transrectalprostate biopsy: a national survey to all urology clinics in Sweden. F1000Research 2020
- 167-170, the information presented is not correct. The EAU guidelines on prostate cancer - ``Ultrasound (US)-guided biopsy is now the standard of care. Prostate biopsy is performed by either the transrectal or transperineal approach``.
Thanks for the advice. We rechecked the EAU recommendation 2021 That are recommending the transperineal technique as the gold standard
Please include the limitations of the study.
Thanks for your suggestions. We formulated the paragraphs. We have not included the UTIs in the because in most cases urinary infections were treated by the GP, we could not trace the data of interest. This must be considered the limitation of our study.
Materials and methods
- 180 Please specify the centers
We specified the centers: Palermo, Pisa, Cuneo
- 183 needs a reference
We added reference 34
describe clearly the dose of antibiotics, mode, and time of administration
We described mode and time of antibiotics administration in the manuscript I 322, 323, 324
- 213-216 hard to understand, please describe ``a rigid scheme of antibiotic prophylaxis``
We added the scheme I 322 – 323
Please specify if the patients had enema the night before or on the morning of the procedure
Specified: I. 324 -325
Conclusions need to be revised
``In diabetic patients the different pharmacological efficacy against the infectious risk is probably caused by the different pathogens pharmacological resistance.`` is based on what results??
Thanks for your advice. Because this was not the aim of our study we removed the consideration from the conclusion.
Round 2
Reviewer 2 Report
Dear Authors,
The article presents some improvements, but it is still insufficient. The authors should try to address all the suggestions from my previous report, not only insert a few sentences. The manuscript is not well written and organized, and the English need significant improvements before it can be processed further. All the sections need to be revised.
Please find other suggestions, and try to address all from my previous:
- The authors should delete from title ``infective`` and leave only complications because they do not present data regarding UTIs
- l. 68-69. The references are still inadequate. The authors must cite EAU and AUA guidelines.
l. 69-74 needs some references - The authors should present the complications in each antibiotic prophylaxis group.
- l. 136-138 needs reference
- l. 192 ref 31 is not from an Eastern Europe
- l. 195 The EAU guidelines on prostate cancer - ``Ultrasound (US)-guided biopsy is now the standard of care. Prostate biopsy is performed by either the transrectal or transperineal approach``. It is not the gold standard; it is the recommended approach. l. 195-199 needs a reference.
- l. 213 Please delete ``All patients underwent TRUSBx``, the information is presented - l 210-211.
- The following suggestions have not been appropriately addressed by the authors (in the article, the information is not present, as suggested in the cover letter)
1. describe clearly the dose of antibiotics, mode, and time of administration
2. Please specify if the patients had enema the night before or on the morning of the procedure
The title of subsection 4.2 should be modified (actually, it should be before statistical analysis) - Antibiotic prophylaxis, patient preparation, and technique of TRUSBx - l. 256 it is not clear what authors understand by secondary prevention therapies
- l. 259-260 should be deleted
- The Discussion sections must be revised.
- The Reference list should respect Journal requirements.
Author Response
Editor
Lois Liu
Assistant Editor
Dorothy Liu
ANTIBIOTICS
Palermo, March 14th 2022
Manuscript #: ANTIBIOTICS-1597891
Title: COMPARISON BETWEEN FLUOROQUINOLONES AND OTHER ANTIBIOTIC PROPHYLAXIS REGIMENS IN PREVENTING COMPLICATIONS IN PATIENTS UNDERGOING TRANSRECTAL PROSTATE BIOPSY
Dear Editors,
Thanks for considering our manuscript for publication in the ANTIBIOTICS. We appreciated the new comments made by the reviewer and we prepared a revised version of our manuscript based on these comments. Manuscript has undergone English language editing by MDPI as recommended. Please also find a point-by-point “response to reviewer” below. We hope you can find this revised version of our manuscript fully suitable for publication in your esteemed journal.
Kindest regards
Gabriele Tulone, MD
Manuscript has undergone English language editing by MDPI as recommended
- The authors should delete from title ``infective`` and leave only complications because they do not present data regarding UTIs
As recommended we change the Title of the manuscript removing “infective”
- 68-69. The references are still inadequate. The authors must cite EAU and AUA guidelines.
69-74 needs some references
We have added references on the advice on the use of levofloxacin as a prophylaxis of transrectal prostate biopsy. These are the European and American guidelines of 2018. In addition we have also included the EMA note on the restriction of the use of this class of anti-antibiotics (EMEA / H / A-31/1452 and UAE 2021) as indicated in the European guidelines of 2021
- The authors should present the complications in each antibiotic prophylaxis group
Unfortunately we cannot go back to this data of the single complications for a single drug which is certainly useful. In the manuscript we preferred to focus on infectious complications, fever and sepsis. We decide to evaluate the complications related to the diabetes as described in tab. 2. A subsequent and prospective study may also focus on the other complications related to the single antibiotic.
- 136-138 needs reference
As recommended we have included a reference. In the article by Yanshu Jia and Liyan Zhao entitled The antibacterial activity of fluoroquinolone derivatives: An update (2018 - 2021), published in Eur J Med Chem 2021 Nov 15; 224: 113741. doi: 10.1016 / j.ejmech.2021.113741 discusses the increased incidence of quinolone resistance among Gram-positive bacteria (including Enterococcus faecalis / E. faecalis and Enterococcus faecium / E. faecium) and Gram-negative (including Escherichia coli / E. coli, Pseudomonas aeruginosa / P. aeruginosa, and Klebsiella pneumonia / K. pneumonia).
- 192 ref 31 is not from an Eastern Europe
Thank we corrected the answer
- The EAU guidelines on prostate cancer - ``Ultrasound (US)-guided biopsy is now the standard of care. Prostate biopsy is performed by either the transrectal or transperineal approach``. It is not the gold standard; it is the recommended approach. l. 195-199 needs a reference.
Thanks for the advise. It is true that the guidelines recommend the perineal approach and still cannot be considered as the gold standard. We have added a meta-analysis conducted by Xiang et al in which the possible complications related to the two procedures are also mentioned. Particularly in the case of perineal biopsy there is a greater risk of post procedural perineal pain.
- 213 Please delete ``All patients underwent TRUSBx``, the information is presented - l 210-211.
We eliminated the repetition
- The following suggestions have not been appropriately addressed by the authors (in the article, the information is not present, as suggested in the cover letter)
describe clearly the dose of antibiotics, mode, and time of administration
2. Please specify if the patients had enema the night before or on the morning of the procedure
The title of subsection 4.2 should be modified (actually, it should be before statistical analysis) - Antibiotic prophylaxis, patient preparation, and technique of TRUSBx
we insert in the manuscript a table with type, mode and time of administration of antibiotic prophylaxis
Type of antibiotc |
Mode of administration |
Time of administration |
Cefixime 400 mg |
Oral |
The night before and the morning of the procedure |
Ceftriaxone 1 g / Fosfomycin 3 g |
Fosfomycin: Oral administrationCeftriaxone: Administration within a muscle. |
Fosfomycin: The night before and the morning of the procedureCeftriaxone: one shot before the biopsy |
Trimethoprim 160 mg /Sulfamethoxazole 800 mg |
Oral |
The night before and the morning of the procedure |
Ciprofloxacin 500 mg |
Oral |
The night before and the morning of the procedure |
Levofloxacin 500 mg |
Oral |
The night before and the morning of the procedure |
Prulifloxacin 600 mg |
Oral |
The night before and the morning of the procedure |
We fixed rectal cleansing with enema. All patients had enema the night before and the morning of the procedure
We modified the title of subsection 4.2 as advised.
- 256 it is not clear what authors understand by secondary prevention therapies
The sentence is for patients with positive pre-procedure urine culture. They performed targeted therapy
- 259-260 should be deleted
We deleted the answer
- The Discussion sections must be revised.
We revised and corrected the discussion by integrating the suggestions made by the reviewers, making the text more organic and complete also thanks to the new references that give greater strength to the text. In addition, a English language editing by the MDPI corrected the errors present and made the text more fluent.
- The Reference list should respect Journal requirements.
We correct the references respecting journal requirements